DOI: 10.1038/s41467-018-06460-2　　OPEN

# Damage-induced reactive oxygen species enable zebrafish tail regeneration by repositioning of Hedgehog expressing cells

Maria Montserrat Garcia Romero [1], Gareth McCathie[1], Philip Jankun[1] & Henry Hamilton Roehl[1]

Many aquatic vertebrates have a remarkable ability to regenerate limbs and tails after amputation. Previous studies indicate that reactive oxygen species (ROS) signalling initiates regeneration, but the mechanism by which this takes place is poorly understood. Developmental signalling pathways have been shown to have proregenerative roles in many systems. However, whether these are playing roles that are specific to regeneration, or are simply recapitulating their developmental functions is unclear. Here, we analyse zebrafish larval tail regeneration and find evidence that ROS released upon wounding cause repositioning of notochord cells to the damage site. These cells secrete Hedgehog ligands that are required for regeneration. Hedgehog signalling is not required for normal tail development suggesting that it has a regeneration-specific role. Our results provide a model for how ROS initiate tail regeneration, and indicate that developmental signalling pathways can play regenerative functions that are not directly related to their developmental roles.

[1] Bateson Centre, Department of Biomedical Sciences, The University of Sheffield, Firth Court, Western Bank, Sheffield S10 2TN, UK. These authors contributed equally: Maria Montserrat Garcia Romero, Gareth McCathie. Correspondence and requests for materials should be addressed to H.H.R. (email: h.roehl@sheffield.ac.uk)

The study of regenerative biology aims to understand the mechanisms and limitations of endogenous regenerative capacity. During vertebrate appendage regeneration in salamander limb, mouse digit, and *Xenopus* tadpole tail and zebrafish fin, a conserved sequence of events takes place after tissue is removed[1,2]. Immediately following tissue damage the wound is closed by active movement of the surrounding epithelia. Next, the epithelial covering of the wound thickens to form what is called the wound epithelia and cells migrate under the wound epithelia to form a tightly packed mass proliferative cells called the blastema. During the final phase of regeneration the blastemal cells and the wound epithelium proliferate and differentiate to give rise to the missing tissue.

How wounding initiates regeneration is a central question in regenerative biology. Tissue damage results in the immediate release of signals such as calcium, ATP, and reactive oxygen species (ROS) which act to stimulate wound closure and the immune response thus limiting detrimental effects of injury[3]. Of these initial signals, ROS stand out as a good candidate for the activation of regeneration. Upon wounding, calcium release results in a rapid burst of ROS that is likely to involve Duox, an NADPH Oxidase[3]. Then ROS, primarily in the relatively stable form of hydrogen peroxide, are thought to diffuse into the neighbouring tissue to act as a paracrine signal (referred to as ROS signalling). ROS exert their effects through the reversible oxidation of cysteine residues in key regulatory proteins[4]. Although precisely how ROS signalling is able to confer specific cellular responses is still poorly understood, studies focused upon the MAPK and Wnt pathways suggest that ROS levels may serve to modulate the activity of diverse signalling pathways[5,6]. These studies indicate that members of the thioredoxin family of redox sensors bind signalling pathway components in a ROS-dependent manner.

The importance of ROS signalling to vertebrate regenerative biology has only recently become apparent. Research on zebrafish larvae has shown that ROS are required for axonal regeneration[7] and that ROS act via a Src Family Kinase (SFK) to promote regeneration of the fin fold[8]. Other zebrafish studies have shown that adult heart and fin require ROS to regrow after wounding[9,10]. ROS have also been shown to act during Xenopus tadpole regeneration[11]. Thus ROS signalling may act more generally as signal that serves to both coordinate the damage response and to initiate the regeneration program.

Another crucial question is how the regeneration of an organ differs from the initial development of that organ[12,13]. For example, the mesenchyme of the developing limb bud resembles the blastema of the amputated limb in both morphology and its expression of *msx* genes. Similarly, the apical ectodermal ridge of the limb bud resembles the wound epithelium and both structures express *dlx* genes. This suggests that once these structures have appeared, regeneration follows the previously established developmental program. Developmental signalling pathways such as Wnt/β-Catenin, FGF, Hedgehog, Retinoic Acid (RA), Notch and BMP have been shown to play important roles during regeneration. We must therefore ask whether the regenerative roles of these pathways are unique to regeneration or are simply a recapitulation of earlier developmental roles.

To shed light on these questions, we have analysed how developmental pathways direct zebrafish larval tail regeneration. Here we provide evidence that Hedgehog signalling is a key regulator of tail regeneration, acting upstream of the Wnt/β-Catenin, FGF and RA signalling pathways. This finding is surprising given that Hedgehog signalling does not play a role in tail development. In addition, we propose that the source of the regenerative Hedgehog signal is notochord cells that are rapidly repositioned to the stump immediately following wounding. Our data suggest that this movement is dependent upon release of ROS from the wound site and requires SFK activity and microtubule polymerisation. Together these data suggest a model that ROS signalling initiates tail regeneration by relocating Hedgehog expressing notochord cells to the wound site.

## Results

**Overview of tail regeneration.** Regeneration in zebrafish larvae has been studied in two contexts: fin fold excision and tail excision[14–17]. During fin fold excision, tissue removal is limited to epithelium and fin mesenchymal cells in the caudal region of the tail (Supplementary Figure 1a). On the other hand, tail excision involves partial removal of neural tube, notochord, muscle, pigment cells, blood vessels as well as the caudal fin fold (Supplementary Figure 1a). Within minutes after tail excision notochord cells move out of the notochord sheath to give rise to a cluster of cells (the "notochord bead") that sit on the stump of the tail (Supplementary Movie 1). Formation of the notochord bead appears to be caused by contraction of the anterior/posterior body axis resulting in pressure build up in the notochord (Supplementary Figure 2). To determine the timing of regeneration after tail excision we examined the expression markers of the different stages during regeneration (Supplementary Figure 1b). By 24 h post excision (hpe), *dlx5a* expression marks the forming wound epithelium, and by 48 hpe the blastema is marked by strong expression of *msxc*. A previous study of fin fold regeneration found that the early blastema is marked by the RA synthesis gene *raldh2* and that RA signalling is required for regeneration[18]. Consistent with this, we found that *raldh2* is upregulated in the forming blastema at 24 hpe. Increased expression of the muscle differentiation gene *myod* is seen between 48 and 81 hpe suggesting that regrowth takes place during this interval. Although these genes are expressed during tail development, they are not detected immediately prior to excision (Supplementary Figure 1b). This indicates that tail excision reactivates expression of these genes. If operated fish are raised past larval development, they appear morphologically normal. However, skeletal visualisation reveals that the internal structure of the tail is modified perhaps due to a defect in notochord extension (Supplementary Figure 1c).

**Developmental signalling pathways.** To begin to understand how developmental signalling coordinates regeneration, we first focused on the FGF pathway to identify ligands and downstream targets that are induced by tissue removal. Similar to previous studies that identified *fgf20a* as a damage-induced ligand in the fin fold[19], we found that *fgf10a* transcripts are upregulated at 24 hpe in cells adjacent to the extruded notochord bead (Fig. 1a). We also found that the target gene *pea3* is induced in the surrounding cells, indicating that these cells are responding to FGF signalling at this time (Fig. 1a). Neither gene is expressed strongly in unoperated tails at this time (Supplementary Figure 3a, b). The allele *tbvbo* encodes a null mutation in *fgf10a* that does not have a discernable effect on normal tail development (Supplementary Figure 4)[20]. Consistent with a role for FGF signalling in larval tail regeneration we found that *fgf10a*[−/−] larvae have reduced regenerative capacity (Fig. 1b). To determine whether the wound epithelium and blastema form properly, we tested levels of expression of *dlx5a* and *raldh2* in *fgf10a*[−/−] larvae and in fish treated with the FGF receptor inhibitor SU5402[21]. Both conditions do not show an observable difference in expression suggesting that FGF signalling does not play a role in the initial patterning of the regenerating tail (Fig. 1c). A previous study found that FGF signalling is required for regeneration-specific proliferation in the fin fold[15], and similarly we found that the

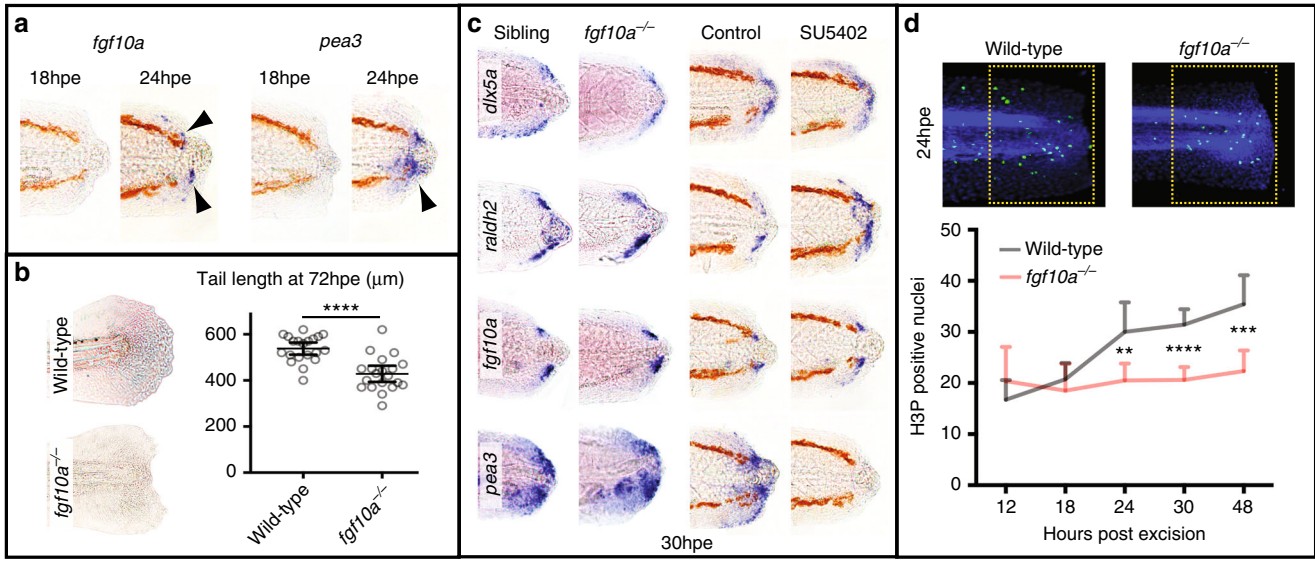

**Fig. 1** FGF signalling is required for cell proliferation during larval tail regeneration. **a** Arrowheads point to expression of the FGF ligand, *fgf10a*, and the downstream target, *pea3*, become evident by 24 hpe (10/10 for both, number of experiments = 3). **b** *fgf10a*−/− fish have reduced tail regrowth compared to wild-type fish. Larvae were imaged at 72 hpe. Tail length was determined using the Measure macro in ImageJ software by placing a rectangle parallel to the body that started at the anus and finished at the caudal end of the fin fold. Unpaired *t* test two-tailed results in *P* < 0.0001 indicated as **** (sibling *n* = 20, *fgf10a*−/− *n* = 20, number of experiments = 3). **c** Expression of *dlx5a* and *raldh2* are not strongly affected when FGF signalling is reduced. SU5402 treatment was done from 26 to 30 hpe at 10 μM concentration. This treatment is sufficient to abolish detection of the FGF downstream target *pea3* (10/10 for each panel, number of experiments = 2). **d** Whereas in wild-type fish DNA mitosis increases after tail excision, in *fgf10a*−/− fish mitosis remains constant (wild type/12 hpe *n* = 8, wild type/18 hpe *n* = 8, wild type/24 hpe *n* = 19, wild type/30 hpe *n* = 21, wild type/48 hpe *n* = 14, *fgf10a*−/−/12 hpe *n* = 8, *fgf10a*−/−/18 hpe *n* = 8, *fgf10a*−/−/24 hpe *n* = 12, *fgf10a*−/−/30 hpe *n* = 20, *fgf10a*−/−/48 hpe *n* = 12, number of experiments = 2). Statistics performed with ordinary one-way ANOVA (nonparametric) with multiple comparisons and Sidak hypothesis testing. ****$P$ < 0.0001, ***$P$ = 0.0001 and **$P$ = 0.005

number of proliferating cells in *fgf10a*−/− larvae is reduced compared to wild-type fish (Fig. 1d). These results suggest that FGF signalling is required for damage-induced proliferation after tail excision.

We next investigated the role of Wnt/β-Catenin signalling during tail regeneration by screening for Wnt genes that are activated after excision. We found that *wnt10a* is upregulated starting at 18 hpe in cells neighbouring the notochord bead and the Wnt/β-Catenin downstream target gene *tcf7* is upregulated also in this region (Fig. 2a). Neither gene is expressed strongly in unoperated tails at this time (Supplementary Figure 3c, d). To further investigate we used the Wnt/β-Catenin pathway inhibitor IWR-1[22] to turn off signalling following excision. Treatment with IWR-1 results in the absence of tail regrowth suggesting that Wnt/β-Catenin signalling is required for regeneration (Fig. 2b). To determine whether *dlx5a* or *raldh2* expression depend upon the Wnt/β-Catenin pathway, we used a heatshock inducible *dickkopf-1b* transgene (*hsp70l:dkk1b-GFP*)[23] to inhibit signalling and a glycogen synthase kinase antagonist (GskXV) to activate signalling[24]. We found that whereas inhibition abolishes expression of both genes, pathway activation leads to expansion of the expression domains of *dlx5a* and *raldh2* (Fig. 2c). Since both FGF and Wnt/β-Catenin pathways appear to act at the same time (24 −48 hpe), we wondered whether they might regulate each other's activity. By manipulation of both pathways we found that each pathway has a positive effect on the other pathway's activity (Fig. 2d, e). These data indicate that Wnt/β-Catenin signalling patterns the early regenerating tail and interacts with FGF signalling.

The third developmental pathway that we have investigated is the Hedgehog signalling pathway. We first tested whether the Hedgehog pathway influences regeneration using the inhibitor cyclopamine[25] (Fig. 3a, b). Cyclopamine treatment following tail

excision results in loss of regeneration and loss of *dlx5a* and *msxc* expression. Consistent with an early role for Hedgehog signalling we found that treatment from −12 hpe to 12 hpe is sufficient to block regeneration. To control for off-target effects of the cyclopamine treatment, we used another Hedgehog pathway inhibitor, LDE225[26], and obtained similar results (Fig. 3a, b). To determine how Hedgehog signalling controls regeneration, we tested for expression of Wnt, FGF and RA genes after cyclopamine treatment and found that their expression is abolished (Fig. 3c). Consistent with Hedgehog signalling acting upstream of FGF signalling we found that cyclopamine treatment strongly reduces proliferation after tail excision (Fig. 3d). Given the early timing of the Hedgehog signalling requirement, it is possible that this pathway acts upstream of the other developmental pathways. To test this, we performed chemical epistasis by treating regenerating fish with cyclopamine to block the Hedgehog pathway while activating the Wnt/β-Catenin pathway with GskXV. We found that GskXV treatment is sufficient to restore expression of the regeneration markers *dlx5a*, *msxc* and *raldh2* indicating that the Hedgehog and Wnt/β-Catenin pathways form a linear pathway (Fig. 3e). Together these data present a model that the Hedgehog pathway plays a key role during regeneration by activating the Wnt/β-Catenin, FGF and RA pathways.

To determine the source and timing of Hedgehog signalling, we looked at the expression of Hedgehog ligands as well as the downstream target *patched1* (*ptch1*). We found that two Hedgehog ligands (*sonic hedgehog a*, *shha* and *indian hedgehog b*, *ihhb*) are strongly expressed in the notochord bead (Fig. 4a and Supplementary Figure 5a). Although this expression appears to be limited to the cells in the bead, this restricted detection is an artefact of the wholemount in situ hybridisation method: When fixed larvae are cut along the coronal axis or obliquely to reveal the notochord prior to hybridisation, then expression of *ihhb* is

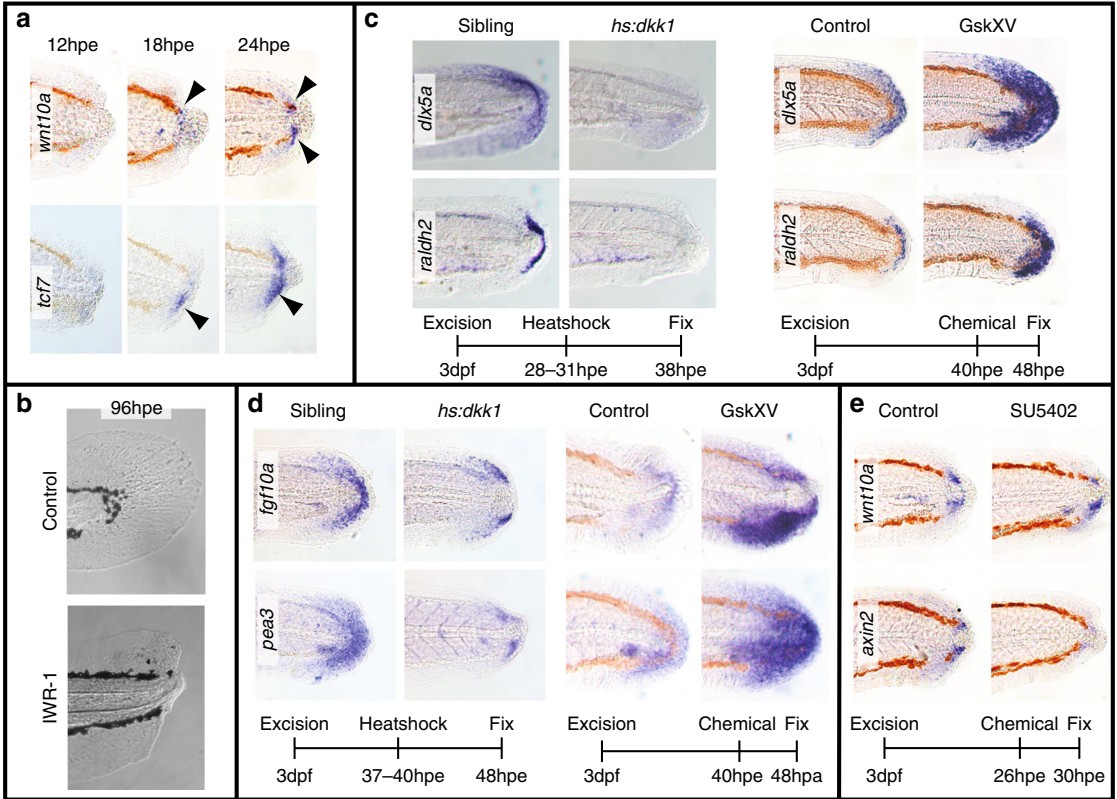

**Fig. 2** Wnt/β-Catenin signalling is required for patterning during larval regeneration. **a** Expression of the Wnt ligand, *wnt10a*, and the downstream target, *tcf7*, become evident by 18 hpe (arrowheads) (*wnt10a*/18 hpe 7/8, *wnt10a*/24 hpe 10/10, *tcf7*/18 hpe 9/10, *tcf7*/24 hpe 10/10, number of experiments = 2). **b** Larvae treated with 10 μM IWR-1 continuously after excision show no signs of regeneration at 96 hpe (10/10 number of experiments = 2). **c** Heterozygous *hs:dkk1* fish and their siblings were heatshocked by placing in an incubator at 39 °C and sorted based by fluorescence present in the transgenic line. Nonfluorescent fish served as control fish. The GskXV treatments were done at 10 μM (*hs:dkk1*/*dlx5a* 10/10, *hs:dkk1*/*raldh2* 10/10, number of experiments = 2) (GskXV/*dlx5a* 10/10, GskXV/*raldh2* 10/10, number of experiments = 4). **d** Expression of *fgf10a* and *pea3* are downregulated in *hs:dkk1* larvae (treatment as in panel **c**) and upregulated in fish treated with 12.5 μM GskXV (*hs:dkk1*10/12 control, 28/30 transgenic, number of experiments = 2) (GskXV control 12/12, treated 11/12 number of experiments = 2). **e** Expression of the Wnt/β-Catenin target gene *axin2* is reduced in fish treated with 10 μM SU5402 (12/12, number of experiments = 3)

detected in the notochord before and after excision (Fig. 4b). This artefact is likely to be because notochord sheath cells deposit a dense extracellular matrix that may restrict penetration of components during in situ hybridisation. Consistent with this model, we found that there is a low level of expression of *ptch1* around the caudal tip of the notochord prior to excision (Supplementary Figure 3f), and high-level expression after the notochord bead has formed (Supplementary Figure 5a). To further test whether Hedgehog signalling acts upstream of other developmental pathways, we looked to see whether manipulation of any of these pathways affects the wound-induced expression of *ptch1* and found that none of the treatments alter its expression (Supplementary Figure 5b). Together these data suggest the model that formation of the notochord bead provides for a new source of Hedgehog signalling that acts upstream of other developmental pathways.

In the fin fold excision model of regeneration, the notochord is not injured suggesting that the initiation of regeneration may take place by a mechanism that does not rely upon the notochord bead and Hedgehog signalling. To test this we treated fish with cyclopamine after fin fold excision. We found a mild effect in that the fin fold regenerates at a slower rate when Hedgehog signalling is blocked (Supplementary Figure 6a). Consistent with this we do not detect upregulation of *ihhb* or *ptch1* after fin fold excision (Supplementary Figure 6b). Although these results suggest that the Hedgehog pathway may act during fin fold regeneration, they

indicate that it is unlikely to play the same crucial role that it does during tail regeneration.

**ROS activate Hedgehog signalling during tail regeneration.** Given that ROS signalling has been proposed to activate regeneration in different models, it is possible that ROS initiate larval tail regeneration upstream of Hedgehog signalling. High levels of ROS are synthesised in cells along the edge of the stump starting immediately after excision (Fig. 5a). To begin to test the role of ROS we blocked their generation with the flavoenzyme inhibitor diphenylene iodonium (DPI) which inhibits NADPH oxidases[3]. Treatment with 150 μM DPI from 1 h prior to excision until 1 hpe is sufficient to reduce regeneration by 50% (Fig. 5b, c) and strongly reduces wound-induced ROS levels (Fig. 5a, d). DPI may affect other signalling pathways besides ROS that involve flavoenzymes or indeed may have unrelated off-target effects[27]. To control for these potential effects of the DPI treatment, we utilised the ROS scavenger MCI186[11] which also reduces ROS levels and regenerated tail length (Fig. 5d, e). The effect of MCI186 is milder than that of DPI but increasing levels of MCI186 results in toxicity, so we kept the levels of MCI186 below this threshold. Consistent with ROS signalling activating regeneration upstream of the Hedgehog pathway, we found that wound-induced expression of *ptch1*, *raldh2*, *tcf7* and *pea3* is reduced following DPI treatment (Fig. 5f).

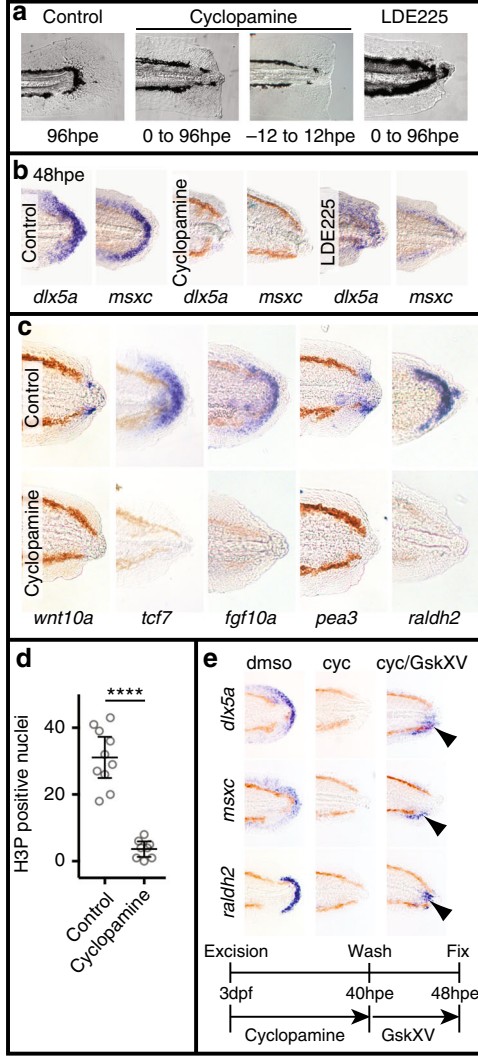

**Fig. 3** Hedgehog signalling plays a central role during larval tail regeneration. **a** Continuous treatment with the Hedgehog pathway inhibitors cyclopamine (20 μM) and LDE225(20 μM) blocks regeneration (Cyclopamine 10/10, LDE 10/10, number of experiments = 2) (Control 20/20, number of experiments = 4). Cyclopamine pulse treatment blocks regeneration (8/10, number of experiments = 2). **b** Larvae treated with cyclopamine (20 μM) and LDE225 (20 μM) from 0 to 48 hpe have reduced expression of dlx5a and msxc (for all panels results are 10/10, number of experiments = 2). **c** Wound-induced expression of markers for the Wnt/β-Catenin, FGF and RA pathways is lost after treatment with cyclopamine. Larvae in the wnt10a and pea3 panels were treated with 50 μM cyclopamine from 0 to 30 hpe. tcf7, fgf10a and raldh2 panels were treated with 20 μM cyclopamine from 0 to 48 hpe (for all panels results are 10/10, number of experiments = 2). **d** Cell proliferation is inhibited by treatment with cyclopamine (control n = 10, cyclopamine n = 8, number of experiments = 2). This analysis was done as in Fig. 1d. Unpaired t test two-tailed results in P < 0.0001 indicated as ****. **e** Upregulation of Wnt/β-Catenin signalling after cyclopamine treatment restores expression of dlx5a, msxc and raldh2 (for all panels results are 10/10, number of experiments = 3). Arrowheads indicate partial rescue of marker expression. Larvae were treated from 0 to 40 hpe with 50 μM cyclopamine and then incubated in 10 μM GskXV until fixation

We next sought to determine whether ROS signalling potentially regulates regeneration by promoting the formation of the notochord bead. When we measured the size of the notochord bead at 4 hpe, we found that DPI treatment reduces

bead formation, but that MCI186 treatment does not (Fig. 6a). As SFKs have been proposed to act downstream of ROS signalling during fin fold regeneration, we tested whether the SFK inhibitor PP2[28] influences bead size. We also used nocodazole that interferes with microtubule polymerisation in an attempt to block bead formation by an independent mechanism. We found that both compounds have a strong effect on bead formation (Fig. 6a). To assess how these inhibitors affect wound-induced activation of Hedgehog pathway, we analysed levels of ptch1 and ihhb expression. We found that these compounds reduce the amount of ihhb transcripts in the notochord bead (Fig. 6b, c) and reduce ptch1 to levels similar to those seen with DPI treatment (Supplementary Figure 7). Although nocodozole treatment would also be predicted to affect ciliogenesis and thus Hedgehog signal transduction, the observation that treatment blocks bead formation and Hedgehog ligand presentation suggest that nocodozole exerts its effect upstream of ciliogenesis. Interestingly, DPI treatment can result in the loss of ihhb expression even when the bead is fully extruded suggesting that a burst of ROS production may be required for both notochord extrusion and expression of ihhb in the notochord bead (Supplementary Figure 8). Together these data suggest that ROS/SFK-dependent bead formation is a necessary step in the regeneration of the tail.

To further control for potential unintended effects of chemical treatments we decided to test whether we could rescue DPI and PP2 treatment by activating Wnt/β-Catenin signalling as we have done for cyclopamine treatments (Fig. 3e). We treated fish at the time of excision with either DPI or PP2 to inhibit bead formation, allowed the fish to recover and then the next day activated WNT/β-Catenin signalling with GskXV. Rescue was quantified by measuring the area of raldh2 expression after RNA in situ analysis. Both DPI- and PP2-treated fish show a significant level of recovery of raldh2 expression after treatment with GskXV (Supplementary Figure 9). This experiment suggests that the effects of DPI and PP2 are not simply due to toxic off-target effects and support the hypothesis that ROS and SFKs act upstream of Wnt/β-Catenin signalling. However, given the limitations of chemical inhibition specificity, these treatments should be interpreted with caution and further analysis of ROS signalling is needed to confirm its role in regeneration.

Having found that DPI and PP2 both strongly reduce notochord bead extrusion, we decided to test whether these compounds affect early cell shape changes seen directly after tail excision (Supplementary Movie 1). Within the first few minutes after excision there is a change in the curvature of the cell membranes which initially bow towards the anterior, then change to bow towards the posterior (Supplementary Figure 2). This change suggests that these cells are passively being forced towards the open end of the notochord perhaps due to increased pressure within the notochord. To quantify this change we chose to measure the Menger curvature which is the inverse of the radius of a circle that approximates the curved arc of the cell membrane. A cell membrane that runs perpendicular to the notochord sheath will result in a Menger curvature of 0, one bowed to the posterior results in a positive value and one bowed to the anterior a negative value. We measured changes in notochord cell curvature during the first 20 min after excision and found that while control fish show a dramatic change in curvature, those treated with DPI or PP2 resemble uncut fish (Supplementary Figure 10).

A second early morphological change after tail excision is contraction of the trunk along the anterior/posterior axis (Supplementary Movie 1). Contraction along this axis could result in increased pressure within the notochord that may result in expulsion of cells from the open end of notochord. We measured trunk length of individual animals before excision and then measured the same animal again at 2 hpe (Supplementary

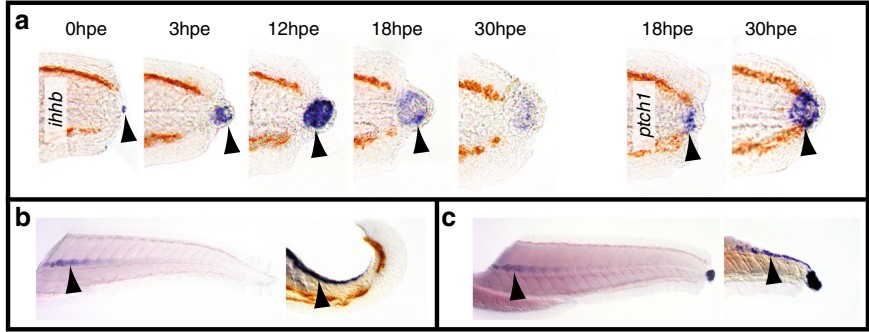

**Fig. 4** The Hedgehog pathway is activated immediately after tail excision. **a** *ihhb* expression is detected within the notochord bead and continues until 30 hpe (0 hpe 2/10, more than 8/10 for the other time points, number of experiments = 3). Expression of *ptch1* is upregulated by 18 hpe and continues after 30 hpe (10/10 for both time points, *n* = 2). Arrowheads point to expression domains. **b** Oblique and coronal sections reveal that *ihhb* is expressed in the notochord before (12/14, number of experiments = 2). **c** *ihhb* is expressed in the notochord after tail excision (7/7, number of experiments = 2)

Figure 11). We found that whereas untreated fish contract by an average of 4.4%, DPI and PP2 only contract by 1.7 and 1.2% respectively. Taken together these data suggest that ROS/SFK signalling is required for early cell movements immediately following tail excision.

**ROS and Hedgehog signalling do not have similar roles during tail development.** Given the complex signalling interactions that take place during regeneration, we wondered whether these interactions are also required during normal tail development. Although interactions between the FGF, Wnt/β-Catenin and RA pathways are crucial for tail formation in chick, mouse and zebrafish[29–32], roles for ROS and Hedgehog signalling have not been described. To test whether ROS act during tail development, we treated larvae with DPI during axis elongation and found that morphologically tails are unaffected by this treatment (Fig. 7a). Likewise, expression of Hedgehog ligands is not affected by DPI treatment during tail development (Fig. 7b). To functionally test whether Hedgehog signalling acts on the tail bud, we treated fish with cyclopamine and found that blocking Hedgehog signalling does not affect *raldh2* or *tcf7* levels (Fig. 7c). Consistent with this, we found that cyclopamine-treated fish have relatively normal tail development except for the formation of U-shaped somites due to a known function of Hedgehog signalling in somite patterning (Fig. 7d)[25]. Furthermore, fish carrying null mutations in the Hedgehog receptor gene *smoothened* (*smo*[b577])[33] form a tail, but lack Hedgehog signalling (Fig. 7d). *smoothened* is absolutely required for Hedgehog signalling as it has no paralogues and there is no evidence for its redundancy with other genes. The possibility that low levels of maternal Smoothened activity may play a role in tail development is very unlikely as maternal-zygotic *smoothened* mutants form a tail[34]. These data indicate that proposed regulatory interactions mediated by ROS and Hedgehog do not act during normal tail development and have specific functions during tail regeneration.

**Discussion**
In the course of this study, we have uncovered a mechanism by which damage-induced ROS signalling may initiate regeneration (Fig. 7). Our model does not exclude the involvement of other signals such as TGFβ, EGF (epidermal growth factor), hypoxia and noncanonical WNT signalling which are also likely to play roles in larval tail regeneration[14,23,35,36]. Rather this model is intended to provide a framework for future analysis of larval tail regeneration in fish. Surprisingly our data suggest that ROS influence regeneration by causing the rapid repositioning of

Hedgehog-expressing notochord cells to the site of the wound. Prior to injury, notochord cells express Hedgehog ligands but this expression has little or no direct effect on cells neighbouring the notochord as judged by expression of *ptch1*. Once the sheath is breached and notochord cells are extruded to form a bead, high levels of Hedgehog ligands signal to the surrounding tissue. Another surprising result is that although Hedgehog signalling is required for tail regeneration, it is dispensable for tail development. Thus the Hedgehog pathway plays a regeneration-specific role in this context and acts as a relay between the immediate damage response and the expression of signalling pathways known to coordinate both development and redevelopment of the tail.

The important role played by the notochord extrusion and Hedgehog pathway activation in zebrafish larvae may be conserved during tail regeneration in other aquatic vertebrates such as frogs and salamanders. One study has shown that in *Xenopus* tadpoles a notochord bead forms at the stump within the first few hours of tail excision, and that these extruded notochord cells express Hedgehog ligands from 24 to 48 hpe[37]. This study went on to show that tadpoles treated with cyclopamine immediately following tail excision have reduced regeneration. Another study has shown that ROS signalling is required for induction of FGF and Wnt/β-Catenin signalling at 36 hpe[11]. Based upon this timing, it is possible that Hedgehog signalling from the notochord bead also acts to link ROS signalling to redevelopment of the tadpole tail. The picture from axolotl is intriguing, as although Hedgehog signalling is important for axolotl tail regeneration, the expression of *Sonic hedgehog* is restricted to the floor plate of the neural tube[38]. It would be interesting if the Hedgehog pathway has maintained its role as a key regulator of tail regeneration in axolotl despite its expression being limited to the floor plate. In addition to its role in tail regeneration, Hedgehog ligands *shha*, *ihhb* and *desert hedgehog* have been shown to be upregulated during heart regeneration where they direct epicardial regeneration[39], Hedgehog signalling activates Wnt/β-Catenin genes during newt limb regeneration[40] and regulates axon guidance during nerve regeneration[41]. As these tissues also require ROS production to regenerate, it will be interesting to test whether the Hedgehog pathway acts to link ROS signalling to redevelopment in these contexts.

In our model of tail regeneration, we have placed the FGF, Wnt/β-Catenin and RA pathways in a "redevelopment" module because broadly speaking these pathways interact in a similar ways during development and regeneration (Fig. 8). This makes sense because it is unlikely that organisms would evolve entirely new mechanisms to regrow tissue, when a pre-existing

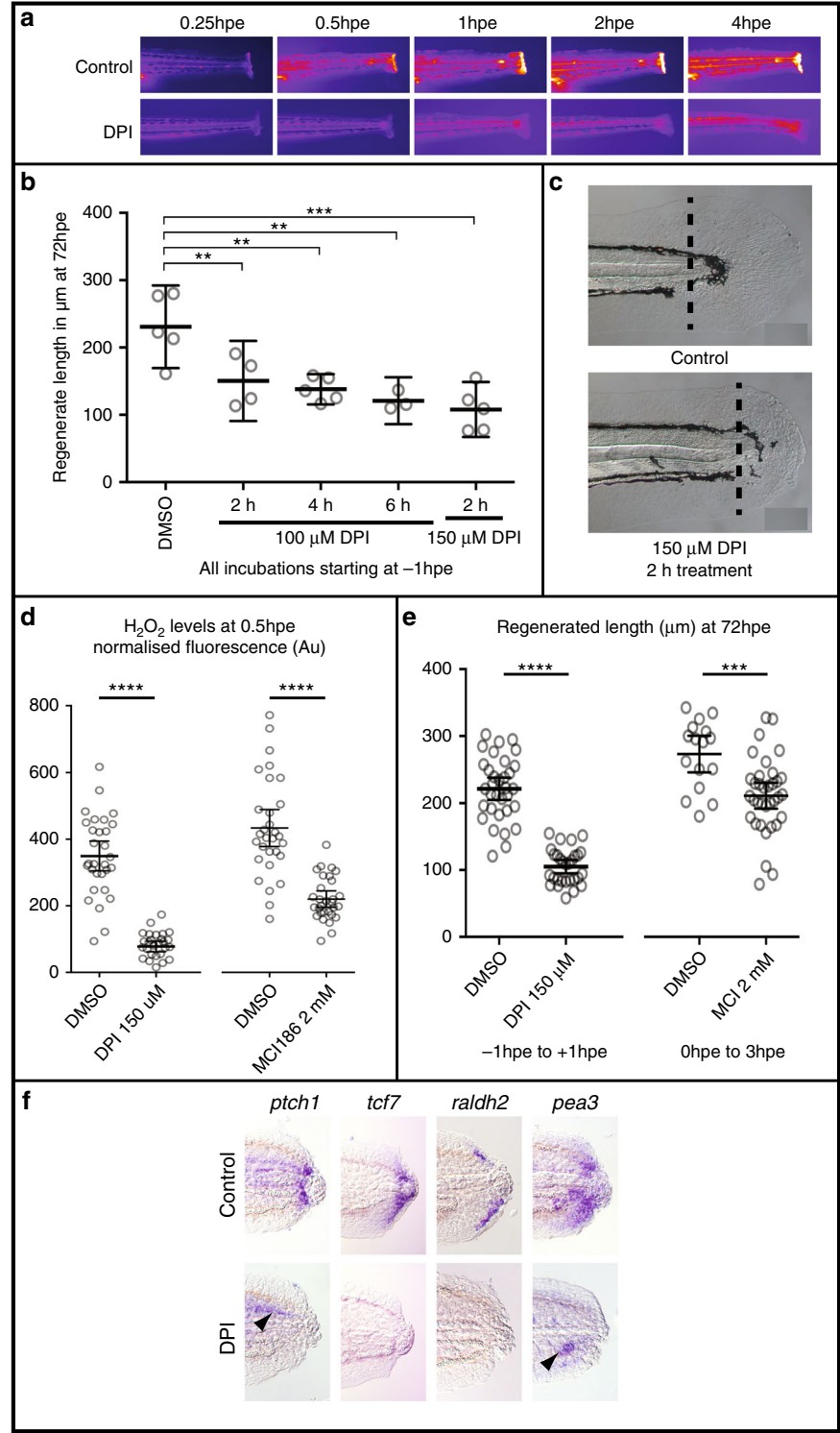

developmental module would suffice. However, this may be an oversimplification as there are some fundamental differences between the origins and movement of precursor cells during development and redevelopment.

In contrast, Hedgehog signalling plays an unexpected role during tail regeneration, raising the question of how Hedgehog signalling has evolved to act upstream of the redevelopment module. If in nature larvae are often losing the end of their tails, then it is reasonable to suggest that Hedgehog regulation of the redevelopment module has evolved due to selective pressure.

Alternatively, Hedgehog signalling may interact with FGF, Wnt/β-Catenin and RA pathways during another developmental process that is unrelated to tail regeneration. In this case its regulatory role could have evolved and be maintained by selection for this unrelated developmental process. This second model does not require selective pressure to evolve Hedgehog's regenerative role, and rather suggests that an existing developmental signalling network has been co-opted to serve during regeneration.

A related question is how ROS/SFK signalling has evolved to regulate notochord extrusion. Here we propose that ROS cause

**Fig. 5** ROS activity immediately after tail excision. **a** Time course showing production of ROS after tail excision is reduced in larvae treated with 150 μM DPI from 1 h prior to excision. Larvae were bathed in 10 μM PFBS-F (205429, Santa Cruz) to detect ROS. **b** Optimisation to show that treatment for as little as 2 h (1 h pretreatment and 1 h post-treatment) is sufficient to reduce tail regrowth by >50%. Regrowth was quantified from images using the Measure macro in ImageJ by placing a rectangle parallel to the body that started at the end of the notochord and finished at the caudal end of the fin fold. Significance was calculated using one-way ANOVA with Dunnett's multiple comparisons test and each sample compared to the DMSO control (DMSO $n = 5$; 100 μM DPI/2 h $n = 4$ and $P = 0.0096$; 100 μM DPI/4 h $n = 5$ and $P = 0.0018$; 100 μM DPI/6 h $n = 3$, $P = 0.0015$; 150 μM DPI/2 h $n = 5$, $P = 0.0001$; number of experiments = 1). **c** Representative larval tails showing the extent of tail regrowth after DPI treatment. **d** Quantification of PFBS-F fluorescence shows that DPI treatment has a stronger effect on ROS levels than MCI186. Thirty larvae were analysed for each sample except for DPI which had only 29 (number of experiments = 2). **** indicates $P < 0.0001$. **e** Comparison of the efficacy of DPI to MCI186 in regards to regenerated tail length. Measurements were made as in panel **a** (DMSO $n = 33$, DPI $n = 29$, number of experiments = 3) (DMSO $n = 16$, MCI186 $n = 33$; number of experiments = 3). **** indicates $P < 0.0001$, *** indicates $P = 0.005$. **f** DPI treatment inhibits wound-induced activation of the Hedgehog, Wntβ-Catenin, RA and FGF pathways. Larvae were pretreated with 150 μM DPI for 1 h and for 6 hpa in 100 μM DPI. Larvae were then washed in E3 buffer and incubated until 24 hpa or 48 hpf in the case of *tcf7* (*ptch1* 19/19, *tcf7* 7/9, *raldh2* 13/19, *pea3* 9/9). Arrowheads point to expression of *pea3* and *ptch1* that is found in unoperated animals (see also Supplementary Figure 2)

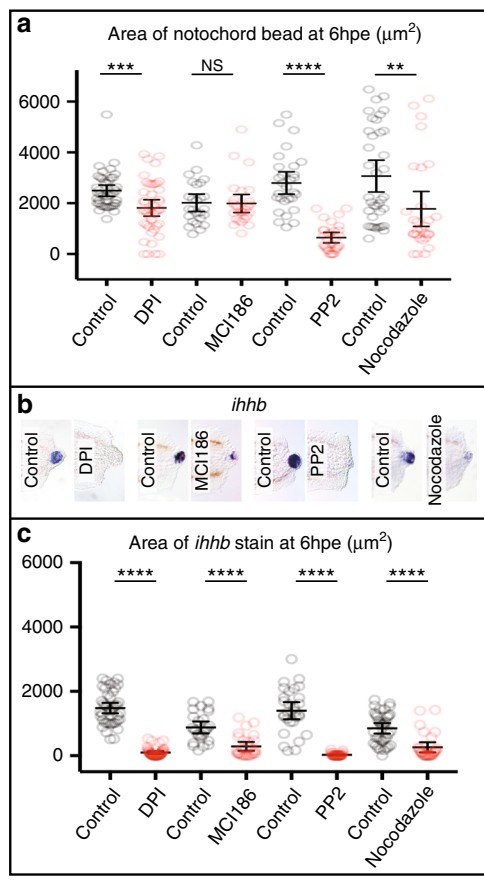

**Fig. 6** DPI, MCI186, PP2 and nocodazole affect notochord bead extrusion and *ihhb* expression. **a** Bead extrusion is affected strongly by PP2 treatment and not by MCI186 treatment. DPI and nocodozole have intermediate effects on extrusion (DMSO $n = 42$, DPI $n = 43$, number of experiments = 3) (DMSO $n = 26$, MCI186 $n = 27$, number of experiments = 3) (DMSO $n = 29$, PP2 $n = 29$, number of experiments = 3) (DMSO $n = 37$, nocodozole $n = 29$, number of experiments = 3). **b** Representative *ihhb* expression patterns after different treatments. **c** Quantification of *ihhb* stain from panel **b** shows that all treatments reduced the level of expression in the notochord bead. DPI (150 μM) treatment started at 1 h prior to wounding and ended at 1 hpa, after which larvae were washed in E3 buffer and incubated until 6 hpe. MCI186 (2 mM) treatment started at 0 hpe and ended at 3 hpe. PP2 (20 μM) treatment started at 1 h prior to wounding and ended at 6 hpe. Nocodazole (10 μg/ml) treatment started at 0 hpe. Area of bead was quantified using the Measure function in ImageJ. Statistics shown are unpaired *t* test two-tailed with **** indicating $P < 0.0001$, ** indicating $P = 0.0066$ and *** indicating $P = 0.0008$

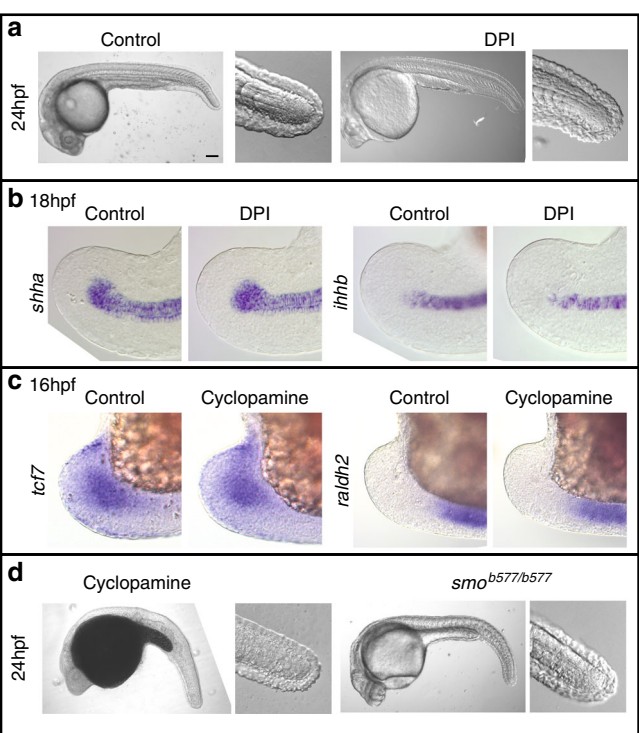

**Fig. 7** ROS and Hedgehog signalling do not have similar roles during tail development. **a** DPI treatment at 100 μM from 14 to 24 hpf does not affect larval tail morphology (10/10, number of experiments = 3). Scale bar is 200 μm. **b** *shha* and *ihhb* expression in the tail is not affected by DPI treatment at 100 μM from 14 to 18 hpf (20/20 for all panels, number of experiments = 2). **c** *tcf7* and *raldh2* expression in the tail bud is not affected by treatment with 20 μM cyclopamine from 8 to 16 hpf (20/20 for all panels, number of experiments = 2). **d** Loss of Hedgehog signalling does not affect tail formation. Larvae were treated with 20 μM cyclopamine from 8 to 10hpf (10/10, number of experiments = 2). Homozygous *smo^{b577}* mutants (7/7, number of experiments = 2) lack zygotic Hedgehog signalling activity. Embryos were manually dechorionated prior to chemical treatment

contraction along the anterior/posterior axis which results a build-up of pressure within the trunk of the fish. Given that the notochord sheath forms a tube-like structure, compaction along the anterior/posterior axis could build up pressure and force notochord cells rapidly out of the open end. The driving force for contraction may be the mass movement of epithelial cells towards the stump that has been described after fin fold excision[42]. The

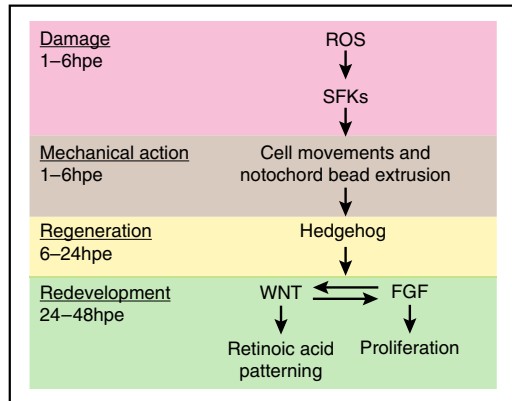

**Fig. 8** A model for tail regeneration. This figure summarises our model for zebrafish larval tail regeneration. On the left side are the different stages of this process with the approximate timing. Please see the text for further details

molecular mechanism for notochord bead extrusion is likely to involve ROS signalling through SFKs as well as microtubule polymerisation. Further evidence for this mechanism comes from a range of models. Studies in *Xenopus* have shown that microtubule polymerisation is required for wound closure[43] and ROS have been shown to play a role in wound closure in *Caenorhabditis elegans*[44] and more recently in zebrafish[45]. Yoo et al. have shown that the SFK Lyn acts as a receptor for ROS signalling[28] and several studies have suggested the model that SFKs directly phosphorylate microtubules to promote their polymerisation and/or stabilisation[46–48]. Thus, it is possible that the notochord bead results from ROS-dependent morphological changes that pressurise the notochord forcing cells out of the open end. Once formed, the notochord bead then acts as a source of Hedgehog signalling to promote tail redevelopment.

## Methods

**General methods.** Experimental procedures and fish maintenance were performed using standard methods[49]. All animal husbandry and experimentation was carried out under the supervision and approval of the Home Office (UK) and the University of Sheffield Ethics Board. Adult zebrafish were maintained with a 14 h light/10 h dark cycle at 28 °C according to standard protocols and were mated using pair mating in individual cross tanks. For more information on how individual experiments were performed, please refer to the figure legends and the sections below. The strains used in this study are *hsp70l:dkk1b-GFP*[23], *fgf10a*[tbvbo20], *smoothened*[b577 33]. All images were taken with anterior to the left and dorsal up.

**Chemical treatment.** All chemical treatments in this study were done in embryo media (E3) at 28.5 °C unless otherwise stated. Control fish were treated with the appropriate solvent. IWR-1 (I0161, Sigma), LDE225 (S2151, Selleckchem), GskXV (361558, Merck), SU5402 (572630, Merck), nocodazole (1228, Tocris Bioscience), DPI (D2926, Sigma) were dissolved in DMSO prior to use. Ethanol was used to solubilise cyclopamine (C4116, Sigma). MCI186 (443300, Merck) was dissolved directly in E3 immediately before use.

**Tail excision.** Fish were anesthetised in 40 μg/ml Tricaine (3-amino benzoic acidethylester) in E3. A scalpel was used to remove the end of the tail using the pigment gap as a reference (Supplementary Figure 12). For short-term treatments (4 hpe or less) larvae remained in tricaine.

**Statistics and animal numbers.** Statistical analysis and the numbers of animals used is reported in the figure legends. To report in situ expression patterns, a representative animal is shown in the figure panel. To score the consistency of the expression, the number of animals in the panel is indicated by a fraction. For example, if the panel shows a lack of expression and the fraction is 9/10, this indicates that nine out of ten animals in that experimental group lacked expression. For experiments that were quantified, graphs were generated and analysed in Prism 7 software. Error bars indicate the 95% confidence interval and the centre bar represents the mean. Individual circles represent individual animal tested. The figure legends indicate the type of statistical test applied, the *P* values and the

number of animals per sample (*n* =). The number of times the experiment was performed is indicated.

**Image analysis.** Images were blinded before analysis using the macro entitled "Renaming images for blind analysis" available as Supplementary Software 1. This macro takes a folder of images and duplicates every image renaming it with a random code. The duplicated images are stored in a separate folder and a table is saved that serves as a key to reveal which number corresponds to which original image. If possible, images are captured so as not to reveal the experimental group. For example *fgf10a*$^{-/-}$ fish lack pectoral fins so it is important that pectoral fins are not included in the images when mutant fish are being compared to wild type.

For quantification of RNA in situ data two macros were applied, one to set the RGB limits for the quantification ("Setting the RGB threshold limits"), and a second to quantify the area of staining ("Quantification of images from preset RGB threshold limits"). These are available as Supplementary Software 3 and Supplementary Software 4, respectively. Briefly, the RGB images were converted into three 8-bit grey scale images representing the red, green and blue channels and each pixel has a value between 0 and 255 based on its intensity in that channel. RGB colour thresholds simply set a maximum and minimum intensity for each of the red, green and blue channels, and then select all pixels which fall within the set ranges for all three channels (Supplementary Figure 13). This allows pixels of a specific colour to be automatically selected based on their RGB intensity values. For this analysis, strong and weak stainings were manually analysed to determine appropriate RGB colour threshold values which defined the area of the blue dye (oxidised BCIP) deposited during the staining procedure. These RGB thresholds were set for each experiment due to the inherent variations in enzymatic staining procedures, combined with differences in probe staining patterns, but remained constant during quantification to allow comparative analysis. Once manually determined by a trial-and-error procedure and visual confirmation, these values for each of the RGB channels are input to the first macro ("Setting the RGB threshold limits") which saves them to a temporary file. The second macro ("Quantification of images from pre-set RGB threshold limits") imports these RGB threshold values from the temporary files and quantifies the number of pixels within the wound area which fall within this RGB threshold. This macro creates a mask which can be manually adjusted to exclude any staining artefacts (e.g. two bubbles seen in the example below). The macro cycles through every image in the target folder and saves a table of staining area using the blinded code names for each file. The user must then use the key file to rename the files to the original designation.

For fluorescence quantification, the $H_2O_2$ signal was quantified in Fiji using the "Wound-induced $H_2O_2$ quantification macro" available as Supplementary Software 2. Briefly, the macro automatically detects the embryo outline, measures the mean fluorescent intensity within 50 μm of the wound edge and the median fluorescence intensity from an area of trunk 1 mm distal to the wound (Supplementary Figure 14). The median fluorescent intensity of the trunk is then subtracted from the mean fluorescent intensity of the wound to control for the basal oxidative state within each embryo. Fish were bathed in E3 medium supplemented with 10 μM pentafluorobenzenesulfonyl fluorescein (Santa Cruz Biotechnology #sc-205429) and at 30 min post excision embryos were imaged under a Zeiss Axio Zoom V16 stereomicroscope with an AxioCam MRm camera and Zen 2 (Blue Edition) software, 2-channel, brightfield and fluorescent (YFP filter: 489-505/516/524-546).

The Menger curvature was measured using circumcircle.ijm, an ImageJ macro kindly provided by Dave Mason, The Centre for Cell Imaging, University of Liverpool. To select each cell, a rectangle was drawn to indicate the distance of 600 μm from the stump, and the closest cell that spanned the width of the notochord was selected for analysis (Supplementary Figure 15). If a cell membrane contacts other notochord cells, the cell membrane becomes bent and these cells cannot be used for analysis. To create a circumcircle, a triangle is made by selecting two points where the cell contacts the notochord sheath and one point on the cell membrane that is the furthest from the axis of these two points. If the circumcentre is anterior to the cell membrane, then the resulting measurement remained positive, if the circumcentre falls posterior to the cell membrane then the curvature is set to negative. This macro is available here: (https://bitbucket.org/davemason/threepointcircumcircle). An adapted version, "Circumcircle", is available as Supplementary Software 5.

For the trunk contraction measurements, fish were imaged immediately prior to tail excision, incubated individually for a further 2 h and imaged again. After blinding, the Measure function of ImageJ was used to determine the contraction of along a distance of approximately eight somites using the positions of somite boundaries and pigment cells as a reference (Supplementary Figure 16). The distance was measured in parallel to the notochord sheath using the rectangle drawing tool.

Quantification of proliferation was done as follows: Fish were stained with Phospho-Histone H3 (ser10) antibody (ThermoFisher) followed by goat anti-rabbit Alexa 488 (green) and imaging was done on an Olympus FV100 microscope. Maximum intensity projections were quantified in Perkin Elmer Volocity software by counting all points greater than 50 μm$^2$ in size and within 500 μm of the caudal end.

**RNA in situ method and probe information**. RNA in situ analysis was done using published protocols and PCR generated probes[50]. Sequences of the probes used in this study are in Supplementary Table 1.

## Data availability

The authors declare that all data supporting the findings of this study are available within the article and its supplementary information files or from the corresponding author upon reasonable request.

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

## Acknowledgements

The authors would like to thank the reviewers for their helpful comments. We would also like to acknowledge the assistance of staff from the Sheffield Zebrafish Aquaria and the Light Microscopy Facility. Supported by grants from Cancer Research UK (C11413/A12714) and Medical Research Council (MR/J001457/1) to H.H.R.

## Author contributions

All authors conceived and performed the experiments, and the manuscript was written by H.H.R. with help from G.M. and M.M.G.R.

## Additional information

**Competing interests:** The authors declare no competing interests.

