## [Peer Review File · Nature Communications]

Reviewers' Comments:

Reviewer #1:

Remarks to the Author:

The study by Romero et al presents results to address two separate questions in the regenerative biology field – what is the role of ROS signaling and does regeneration recapitulate development. Seemingly, one would not design a study to address these very different questions but the authors do a good job of melding their findings. They show that RA, FGF, and WNT signaling are required for fin regeneration and they are enacted relatively late after tail excision. These are not controversial findings as this temporal order and the necessity of these pathways has been established previously in zebrafish and in *Xenopus*, albeit with different models. However, the analysis of HH provided a new wrinkle and the data are convincing that HH operates upstream of RA, FGF, and WNT during regeneration but not tail development. The differences shown between regeneration and tail development represent a novel finding and I enjoyed reading the manuscript and I think it will be of interest to others in the regeneration community. Probably the weakest part of the story concerns the notochord bead as a necessary structure and primary source of the elixir (is it only HH?) that initiates regeneration. The importance of the sheath surrounding the notochord bead as a barrier to HH signaling rests on indirect evidence, and whether or not ROS causes the contraction that happens at the margin of tail amputations in fish and amphibians is debatable. Seemingly there is more to the story of initiating regeneration than establishing a structure that allows HH signaling, or else a regenerative response could be induced in an intact tail by HH supplementation or pathway activation. Maybe this could be better clarified.

Line 40-41: I think it might ruffle a few feathers if you say ROS is the best candidate (hypoxia?), maybe tone this down.

Line 162: Probably too strong to say that HH is a master organizer when other pathways that were not examined (tgfb?).

Line 186: Why does cyclopamine slow/inhibit regeneration in the fin fold excision model?

Line 316: There is no mention/description of the fish lines used....do they have RRID's?

Line 212: Nocodazole would be expected to affect cilia, and cilia are important for HH signaling. This should probably be addressed somehow.

Line 292-294: This sentence is a bit vague, can you add more details about mechanism?

Reviewer #2:

Remarks to the Author:

Romero et al. investigate pathways involved in larval tail regeneration in zebrafish. Using in-situ hybridization and tail regeneration length measurements, the authors aim to dissect the downstream signaling cascade by which wound induced ROS leads to regeneration. They propose that ROS activate regenerative signaling by triggering the mechanical extrusion of a "notochord bead". They propose that hedgehog ligands on this notochord bead trigger regenerative signaling when exposed to the injured tissue. It is further suggested that ROS-induced hedgehog signaling specifically regulates regeneration and not developmental growth, which in their zebrafish model coincides with regeneration.

Overall, the study seems well performed. Some of the published experiments repeat previously published results and are of confirmatory nature (e.g., that DPI inhibits regeneration, etc.). The most novel and interesting aspect of this study is the mechanistic link between ROS, notochord bead extrusion, and hedgehog signaling. This link is potentially intriguing, and should be further

strengthened by orthogonal approaches. Some experimental choices and interpretations require additional explanation.

Major comments:

- It is proposed that notochord bead extrusion is triggered by ROS. However, the extrusion mechanism is only superficially addressed by Fig. S2 and a movie. Fig. S2 is supposed to show pressurization of the notochord. I cannot judge whether bowing of a notochord segment borders is a reliable indicator of pressurization. But even taking this at face value, there is no quantification of notochord movement, and it is not shown whether this movement is actually altered by ROS or microtubules as suggested in Fig. 7 and proposed by the authors. The extrusion mechanism as trigger for regeneration seems to be a major novelty of this study. An assay should be developed that measures notochord movement over different animals and experimental perturbations (ROS & MT inhibition, etc.) to strengthen this part.

- Many central experiments rely on broad spectrum antagonists that likely have massive off target effects. Without orthogonal genetic approaches, the use of DPI or broad-spectrum antioxidants is always problematic. DPI is a flavoprotein inhibitor, and flavoproteins are broadly involved in electron transport reactions throughout the cell. Likewise, the relatively simple chemical structure of the MCI antioxidant gives little reason to trust in a specific antioxidant function. To strengthen these experiments, the authors could show that the effects of DPI or MCI186 are bypassed by external ROS application (e.g., bath application of H₂O₂, or inhibition of endogenous antioxidants) as expected if the observed phenotype is indeed due to ROS, and not e.g., general toxicity. Along these lines, an epistasis experiment involving DPI/MCI186 and GskXV could be reassuring. If the ROS inhibitors act specifically, GskXV should restore expression of regeneration markers in the absence of ROS (i.e., under DPI & MCI treatment). Of note, the epidermal permeability of DPI is unclear. It surely diffuses through a wound, but does it equally well diffuse through an intact epithelium at concentrations required for pathway inhibition during development?

Others:

- Why was the hedgehog mutant line shown in Fig. 6 not used to confirm central cyclopamine results (in Fig. 3)?

- I apologize if I missed this: I could not find a comprehensive methods section that describes exposure and imaging conditions for the in-situ experiments in the manuscript pdf. A comprehensive statistical methods description is also missing. Are graphs representing single or aggregated experiments? What types of t-tests and significance thresholds have been used? For the instances where no quantification was provided ("representative images"), could the complete data sets please be provided. The current level of detail may make reproduction difficult.

- The current title of the paper suggests that the study is about the direct activation mechanism of regeneration by ROS. So, I was somewhat expecting the discovery of a novel ROS-sensor for regeneration. But the ROS sensor function of SFKs has been proposed before. I think the title does not accurately highlight the main novelty within this paper.

Reviewer #3:

Remarks to the Author:

This manuscript investigates the roles of several signaling pathways in zebrafish tail regeneration. Most of the pathways and cellular events that the authors describe were already known to play roles in regeneration; the value of this manuscript is that it investigates the relationship between these processes to develop a coherent model of the chain of events regulating regeneration. Briefly, the authors propose that tail amputation generates ROS, which promotes extrusion of the notochord, which then serves as a source of hedgehog, which activates wnt and fgf signaling to promote differentiation and proliferation, respectively. ROS generation and hh signaling are not required for initial tail development, suggesting that their roles are specific to regeneration.

The manuscript is clearly written and persuasive. Suggestions for revision are enumerated below.

1) The manuscript could benefit from more quantification. Much of the data is presented as

individual images, but it is difficult to judge the variability/strength of the results from just these images. For example, the tail regeneration defects caused by treatment with the wnt inhibitor are not quantified. Similarly, in situs show presumably representative images of gene expression, but it would be nice to have an idea of the reliability of the results. The methods state that in situ stains were scored blind, but these scores do not seem to be reported. Was each experiment only done once, or were staining results repeated on multiple days?

2) Extrusion of the notochord is a central event in the model proposed by the author, suggesting that injuries sparing the notochord will not regenerate as well. Could this be tested directly by creating injuries that spare the notochord, but damage other tissues (e.g. the spinal cord and/or somites)?

3) The Methods section is inadequate. The section does not describe the strains of fish used, how they were raised, how amputations were performed, the in situ protocol, statistical methods, etc.

4) Given the fact that in Wnt signaling is not only required for regeneration but can also promote it (Kawakami et al, 2006), does the wnt signaling activator accelerate tail regeneration?

5) One of the key questions raised by this study is how ROS regulates notochord extrusion. I am confused by the authors' discussion of this issue. How is a presumably mechanical (pressure-driven) process, affected by ROS signaling? Could immune cells, recruited by ROS, play a role in this process? A bit more discussion of this issue could be helpful.

Response to Referees

Reviewer #1 (Remarks to the Author):

The study by Romero et al presents results to address two separate questions in the regenerative biology field – what is the role of ROS signaling and does regeneration recapitulate development. Seemingly, one would not design a study to address these very different questions but the authors do a good job of melding their findings. They show that RA, FGF, and WNT signaling are required for fin regeneration and they are enacted relatively late after tail excision. These are not controversial findings as this temporal order and the necessity of these pathways has been established previously in zebrafish and in *Xenopus*, albeit with different models. However, the analysis of HH provided a new wrinkle and the data are convincing that HH operates upstream of RA, FGF, and WNT during regeneration but not tail development. The differences shown between regeneration and tail development represent a novel finding and I enjoyed reading the manuscript and I think it will be of interest to others in the regeneration community. Probably the weakest part of the story concerns the notochord bead as a necessary structure and primary source of the elixir (is it only HH?) that initiates regeneration.

The evidence for this model is that:

- 1) *if we block bead formation using a different methods then regeneration is blocked*
- 2) *Hedgehog ligands are strongly expressed in the notochord bead*
- 3) *Hedgehog signal transduction takes place in cells that neighbour the bead shortly after bead formation.*
- 4) *Hedgehog signalling is required for regeneration to proceed.*

We feel that there is sufficient evidence to support our model

The importance of the sheath surrounding the notochord bead as a barrier to HH signaling rests on indirect evidence,

We have reworded this sentence. line 296

whether or not ROS causes the contraction that happens at the margin of tail amputations in fish and amphibians is debatable.

We have added analysis that shows that constriction along the anterior/posterior axis occurs at the time that ROS are made, and that PP2 and DPI reduce this constriction. Figure S12.

*We have also cited two papers from *C.elegans* and zebrafish studies in the discussion that have proposed a role for ROS in wound closure:*

Xu S, Chisholm AD. C. elegans epidermal wounding induces a mitochondrial ROS burst that promotes wound repair. Developmental cell 31, 48-60 (2014)

LeBert D, et al. Damage-induced reactive oxygen species regulate vimentin and dynamic collagen-based projections to mediate wound repair. Elife 7, (2018).

Seemingly there is more to the story of initiating regeneration than establishing a structure that allows HH signaling, or else a regenerative response could be induced in an intact tail by HH supplementation or pathway activation. Maybe this could be better clarified.

We have added text in the discussion to emphasise that our model does not exclude the involvement of other pathways. line290

Line 40-41: I think it might ruffle a few feathers if you say ROS is the best candidate (hypoxia?), maybe tone this down.

We have rephrased this sentence. line 35

Line 162: Probably too strong to say that HH is a master organizer when other pathways that were not examined (tgfb?).

We have toned down the phrasing of the role of HH signalling to remove "master organizer" and elsewhere where we used the term "orchestrate" as formally HH may play a permissive role.
73, 177, 365

Line 186: Why does cyclopamine slow/inhibit regeneration in the fin fold excision model?

We have not investigated this phenotype further as the mild expressivity would make it difficult to do definitive experiments.

Line 316: There is no mention/description of the fish lines used....do they have RRID's?

We have used the standardised allele and gene names that are set by the Zebrafish International Resource Center. We have also referenced the original papers that described each line in the text. We have added a table to the methods section.

Line 212: Nocodazole would be expected to affect cilia, and cilia are important for HH signaling. This should probably be addressed somehow.

We have added a sentence to cover this issue. line 218

Line 292-294: This sentence is a bit vague, can you add more details about mechanism?

We have rewritten this paragraph to explain this better. line 327

--

Reviewer #2 (Remarks to the Author):

Romero et al. investigate pathways involved in larval tail regeneration in zebrafish. Using in-situ hybridization and tail regeneration length measurements, the authors aim to dissect the downstream signaling cascade by which wound induced ROS leads to regeneration. They propose that ROS activate regenerative signaling by triggering the mechanical extrusion of a "notochord bead". They propose that hedgehog ligands on this notochord bead trigger regenerative signaling when exposed to the injured tissue. It is further suggested that ROS-induced hedgehog signaling specifically regulates regeneration and not developmental growth, which in their zebrafish model coincides with regeneration.

Overall, the study seems well performed. Some of the published experiments repeat previously published results and are of confirmatory nature (e.g., that DPI inhibits regeneration, etc.). The most novel and interesting aspect of this study is the mechanistic link between ROS, notochord bead extrusion, and hedgehog signaling. This link is potentially intriguing, and should be further strengthened by orthogonal approaches. Some experimental choices and interpretations require additional explanation.

Major comments:

- It is proposed that notochord bead extrusion is triggered by ROS. However, the extrusion mechanism is only superficially addressed by Fig. S2 and a movie. Fig. S2 is supposed to show pressurization of the notochord. I cannot judge whether bowing of a notochord segment borders is a reliable indicator of pressurization. But even taking this at face value, there is no quantification of notochord movement, and it is not shown whether this movement is actually altered by ROS or microtubules as suggested in Fig. 7 and proposed by the authors. The extrusion mechanism as trigger for regeneration seems to be a major novelty of this study. An assay should be developed that measures notochord movement over different animals and experimental perturbations (ROS & MT inhibition, etc.) to strengthen this part.

We have added quantification of bowing of notochord cells (Menger curvature, Fig. S11) and trunk compaction along the anterior/posterior axis (Fig. S12). Both these experiments further strengthen our model that pressurisation of the notochord causes expulsion of notochord cells giving rise to the notochord bead. These experiments also show that both DPI and PP2 block these morphological changes consistent with ROS acting to initiate formation of the notochord bead. We tried to do the same with nocodazole, but the trunk usually becomes misshapen after treatment making it difficult to make precise measurements. We have altered our model figure to de-emphasise the link to microtubules, but have mentioned the potential link and references in the discussion section.

In terms of the mechanics of the formation of the bead, we have been unable to come up with another explanation for how cells could move on mass to the posterior of the notochord. Please consider these two arguments:

Active versus passive movement:

The notochord cells do not change position as they move, so if they were actively migrating posteriorly then they would have to pull themselves along the sides of the notochord tube. In this case how could the membrane become bowed to the posterior? Surely if they dragged themselves using the notochord walls the rest of the cell membrane would if anything lag behind and not take the lead. As the cells exit the notochord they pile up on the end of the stump as rounded cells - they do not resemble actively migrating cells. Based upon these observations it is likely that notochord cell movement is passive.

The origin of the passive force:

It is difficult to imagine what passive force could move the cells out of the notochord other than that of pressure (ie other neighbouring cells are not moving, there is no flow of fluid, etc). Where this pressure originates from is less clear. We have shown that there is shortening of the anterior/posterior axis making this a good candidate. However one could imagine other ways to feasibly generate pressure. For example: wounding activates ion pumps that alter the osmotic pressure within the notochord.

We have ensured that the text wording emphasises that the link between axis shortening and notochord cell movement is only a model at this point.

- Many central experiments rely on broad spectrum antagonists that likely have massive off target effects. Without orthogonal genetic approaches, the use of DPI or broad-spectrum antioxidants is always problematic. DPI is a flavoprotein inhibitor, and flavoproteins are broadly involved in electron transport reactions throughout the cell. Likewise, the relatively simple chemical structure of the MCI antioxidant gives little reason to trust in a specific antioxidant function. To strengthen these experiments, the authors could show that the effects of DPI or MCI186 are bypassed by external ROS application (e.g., bath application of H₂O₂, or inhibition of endogenous antioxidants) as expected if the observed phenotype is indeed due to ROS, and not e.g., general toxicity. Along these lines, an epistasis experiment involving DPI/MCI186 and GskXV could be reassuring. If the ROS inhibitors act specifically, GskXV should restore expression of regeneration markers in the absence of ROS (i.e., under DPI & MCI treatment). Of note, the epidermal permeability of DPI is unclear. It surely diffuses through a wound, but does it equally well diffuse through an intact epithelium at concentrations required for pathway inhibition during development?

We have been able to rescue DPI and PP2 treatment using GskXV and this data is now presented in a new figure (Fig. S10, line 227). We have tried to rescue DPI treatment with exogenous H₂O₂ and have been unsuccessful. This may be because the rescue protocol is not optimal (concentration, timing, H₂O₂ has to be provided in a gradient, perhaps H₂O₂ is not the primary ROS, etc). In terms of specificity, it should also be noted that: our treatments are very brief and early, at the time that ROS is detected. And three independent compounds (PP2, MCI186 and DPI) all have similar effects. Nonetheless we do share this reviewer's view that interpretation should be cautious (and have added words to this effect). line 236

In terms of diffusion, it is very difficult to determine the bioavailability of chemicals in any given tissue. However the treatments at early time points during embryonic development (14-24hpf) are likely to be effective:

- 1) *At 14-24hpf the epidermis has not formed.*
- 2) *DPI is a small aromatic compound that is likely to cross membranes quite well*
- 3) *We treated for 10 hours, much more time than is needed to block regeneration*
- 4) *To block regeneration effectively, one must pre-treat larvae with DPI suggesting that DPI is able to penetrate the epidermis before wounding. This was found during the optimisation of DPI treatment (data not shown).*

Others:

- *Why was the hedgehog mutant line shown in Fig. 6 not used to confirm central cyclopamine results (in Fig. 3)?*

By the time larvae reach 72hpf the smo mutants become unhealthy and morphologically deformed.

• I apologize if I missed this: I could not find a comprehensive methods section that describes exposure and imaging conditions for the in-situ experiments in the manuscript pdf. A comprehensive statistical methods description is also missing. Are graphs representing single or aggregated experiments? What types of t-tests and significance thresholds have been used? For the instances where no quantification was provided (“representative images”), could the complete data sets please be provided. The current level of detail may make reproduction difficult.

This information is now included in the methods section. We feel that there is too much data to provide the complete data sets for all experiments (images alone be over a terabyte). If there is data for specific figures that the reviewer would like to include in full this can be done.

• The current title of the paper suggests that the study is about the direct activation mechanism of regeneration by ROS. So, I was somewhat expecting the discovery of a novel ROS-sensor for regeneration. But the ROS sensor function of SFKs has been proposed before. I think the title does not accurately highlight the main novelty within this paper.

We agree that the title could be improved, but have struggled to formulate an improved title. One to consider is "Damage-induced reactive oxygen species enable regenerative signalling by the rapid repositioning of Hedgehog expressing cells."

Reviewer #3 (Remarks to the Author):

This manuscript investigates the roles of several signaling pathways in zebrafish tail regeneration. Most of the pathways and cellular events that the authors describe were already known to play roles in regeneration; the value of this manuscript is that it investigates the relationship between these processes to develop a coherent model of the chain of events regulating regeneration. Briefly, the authors propose that tail amputation generates ROS, which promote extrusion of the notochord, which then serves as a source of hedgehog, which activates wnt and fgf signaling to promote differentiation and proliferation, respectively. ROS generation and hh signaling are not required for initial tail development, suggesting that their roles are specific to regeneration.

The manuscript is clearly written and persuasive. Suggestions for revision are enumerated below.

1) The manuscript could benefit from more quantification. Much of the data is presented as individual images, but it is difficult to judge the variability/strength of the results from just these images. For example, the tail regeneration defects caused by treatment with the wnt inhibitor are not quantified. Similarly, in situ show presumably representative images of gene expression, but it would be nice to have an idea of the reliability of the results. The methods state that in situ stains were scored blind, but these scores do not seem to be reported. Was each experiment only done once, or were staining results repeated on multiple days?

This information is now included in the methods section.

2) Extrusion of the notochord is a central event in the model proposed by the author, suggesting that injuries sparing the notochord will not regenerate as well. Could this be tested directly by creating injuries that spare the notochord, but damage other tissues (e.g. the spinal cord and/or somites)?

This has been partially tested by fin fold excision (Fig. S6). It is not feasible to cut around the notochord, to make a similar wound that leaves the notochord intact while removing a similar amount of tissue.

3) The Methods section is inadequate. The section does not describe the strains of fish used, how they were raised, how amputations were performed, the in situ protocol, statistical methods, etc.

We have added an expanded Methods Section.

4) Given the fact that in Wnt signaling is not only required for regeneration but can also promote it (Kawakami et al, 2006), does the wnt signaling activator accelerate tail regeneration?

Activation of the WNT pathway leads to deformed tails after regeneration is complete, presumably because it needs to be spatially regulated and it plays a patterning role.

5) One of the key questions raised by this study is how ROS regulates notochord extrusion. I am confused by the authors' discussion of this issue. How is a presumably mechanical (pressure-driven) process, affected by ROS signaling? Could immune cells, recruited by ROS, play a role in this process? A bit more discussion of this issue could be helpful.

Reviewer 2 has the same query, please see the response to their first major comment. As far as immune cells, we have looked into this in detail as part of a separate study. We have not found any link between immune response and notochord extrusion: Genetic ablation of immune cells does not affect extrusion. These findings will form part of another manuscript that analyses the role of inflammation in tail regeneration.

Reviewers' Comments:

Reviewer #1:

Remarks to the Author:

I appreciate the changes that the authors made to the manuscript and enjoyed reading the revised version. I think the paper will influence thinking in the field about ROS signaling and the finding of a role for HH signaling in regeneration but not development is novel.

Reviewer #2:

Remarks to the Author:

Romero et al. present a revised version of their manuscript that addresses many of the formal concerns I had in my previous comments. Some of my technical concerns remain.

I appreciate the authors' effort to better explain their ideas regarding the bead extrusion, and their experiment to support the specificity of the DPI effect.

Regarding the extrusion: The authors counter my question with another question, which I cannot answer since I am not an expert in zebrafish notochord hydraulics. Is there perhaps some relevant literature about this that could professionally underline the authors' speculations, and could be cited?

Regarding the specificity of DPI: I appreciate that the authors now explicitly caution their statements on the use of DPI, but this does not change the fact that most of their findings rely on this, and another quite unspecific compound (MCI).

It is great that the authors added a recovery experiment to consolidate their DPI results. The outcome of the H₂O₂ rescue should be discussed in the paper. In my opinion, the provided recovery experiment is not informative enough. In the current experimental setup, DPI was washed out and the tissue was given time to recover. GskXV still works after that. OK, this shows that DPI does not irreversibly damage the tissue. The authors may show instead that Wnt signaling can be directly activated with GskXV in the presence of DPI. If DPI affects signaling at the ROS level, this should work, since Wnt is downstream of ROS according to the authors' model.

Minor questions:

Figure 3c: Why were different incubation timelines used?

Figure 3e: Is there a no-GskXV control?

Figure 5a, c: y-label missing

Figure 6: DPI is an ion. Charged compounds do not penetrate undamaged fish skin well. Please consider.

Figure S6a: May suggest that both reg. mechanisms equally depend on hedgehog signaling with some mild (2x) quantitative difference. Would this be counter to the author's model?

p3/194: "are" missing between "they" and "detected"

Reviewer #3:

Remarks to the Author:

This manuscript, which describes a cascade of signaling events required for tail regeneration in larval zebrafish, has been improved by revision. The expanded methods and detailed figure legends describing statistical analyses are particularly appreciated. Additional epistasis experiments also add confidence to the model. The study's novel insights are that ROS produced after injury promote extrusion of the notochord; hedgehog ligands expressed in the extruded

notochord bead then activate FGF and Wnt signaling to promote regeneration. These studies raise the question of how ROS promotes notochord extrusion. The experiments here do not answer this question, but it will be an interesting topic for future study, and the authors' speculation about this in the discussion is appreciated.

Minor:

--The new title suggested by the author is an improvement over the previous title

--The paper, while clearly written overall, contains grammatical errors and word omissions in several places (e.g. in the abstract). One particularly confusing instance of this is in Line 180, where I believe the authors meant to say "does not rely", rather than "does rely". The manuscript should be carefully copy edited.

Response to Referees

Reviewer #1 (Remarks to the Author):

I appreciate the changes that the authors made to the manuscript and enjoyed reading the revised version. I think the paper will influence thinking in the field about ROS signaling and the finding of a role for HH signaling in regeneration but not development is novel.

--

Reviewer #2 (Remarks to the Author):

Romero et al. present a revised version of their manuscript that addresses many of the formal concerns I had in my previous comments. Some of my technical concerns remain.

I appreciate the authors' effort to better explain their ideas regarding the bead extrusion, and their experiment to support the specificity of the DPI effect.

Regarding the extrusion: The authors counter my question with another question, which I cannot answer since I am not an expert in zebrafish notochord hydraulics. Is there perhaps some relevant literature about this that could professionally underline the authors' speculations, and could be cited?

To the best of my knowledge the physics of the zebrafish notochord have not been analysed. This would be technically quite difficult because of the size of the larvae. There are GFP pressure sensors but apparently these work at pressures that are greater than physiological levels. Mathematical modelling may be useful, but that is not our area of expertise.

Regarding the specificity of DPI: I appreciate that the authors now explicitly caution their statements on the use of DPI, but this does not change the fact that most of their findings rely on this, and another quite unspecific compound (MCI). It is great that the authors added a recovery experiment to consolidate their DPI results. The outcome of the H₂O₂ rescue should be discussed in the paper.

This is now added

In my opinion, the provided recovery experiment is not informative enough. In the current experimental setup, DPI was washed out and the tissue was given time to recover. GskXV still works after that. OK, this shows that DPI does not irreversibly damage the tissue. The authors may show instead that Wnt signaling can be directly activated with GskXV in the presence of DPI. If DPI affects signaling at the ROS level, this should work, since Wnt is downstream of ROS according to the authors' model.

Our model is that a burst of ROS act early to elicit bead extrusion, and that Wnt signalling acts 24 hours later and is downstream of Hedgehog: ROS -> HH Wnt. Based upon the temporal separation of the ROS and Wnt signalling, doing simultaneous treatment with DPI and the Wnt activator GSKXV does not make sense.

That said, it is well documented that ROS modulate Wnt signalling: the redox sensor Nucleoredoxin binds to Dishevelled under reducing conditions to inhibit Wnt signalling (For review see: Redox regulation of Wnt signalling via Nucleoredoxin. Free Radic Res. 2010 Apr;44(4):379-88)). This has been shown in frogs and in cell culture. In other words, we would expect that if we treated fish with DPI during active Wnt signalling (24hpe), we would see a diminution in Wnt readouts confirming a general role for ROS in the Wnt pathway. Treatment with DPI and GskXV simultaneously, would replicate epistasis results that place Nucleoredoxin and Dishevelled upstream of GSK. These experiments would not be relevant to the extrusion of the notochord bead that occurs immediately after excision and is inhibited by treatment with DPI (-1hpe to 1hpe) as well as PP2 and MCI-186. We have added text to make readers aware of the role of ROS in Wnt signalling (as well as MAPK signalling).

As ROS modulation of WNT signalling is not specific to regeneration, we do not think that this data would add to our study, which is focused upon early signals that initiate regeneration.

We hope that the reviewer will accept that the significant amount of effort needed to generate data to support published studies (or not) would not be time well spent.

Minor questions:

Figure 3c: Why were different incubation timelines used?

The times were chosen so that marker gene expression was strong in untreated samples. This emphasises the complete loss of marker gene expression after treatment.

Figure 3e: Is there a no-GskXV control?

We have added these images

Figure 5a, c: y-label missing

y-label is at the top of the figure, this to use space more effectively.

Figure 6: DPI is an ion. Charged compounds do not penetrate undamaged fish skin well. Please consider.

It is very difficult to follow bioavailability and this is beyond the scope of our paper. Since DPI pretreatment (before wounding) is necessary for it to be effective at 72hpf, and the epidermis has not formed at the time of treatment for Figure 6, we feel that it is on the balance likely that DPI does get into the embryos. There are an abundance of papers that utilise DPI in granulocytes and there it is known that Duox is inside vesicles in the cell. DPI clearly gains entry into the cell but the mechanism is unknown. We would prefer that this figure remain as is.

Figure S6a: May suggest that both reg. mechanisms equally depend on hedgehog signaling with some mild (2x) quantitative difference. Would this be counter to the author's model?

We have reworded the conclusion of this paragraph in the results section.

p3/l94: "are" missing between "they" and "detected"
corrected

--

Reviewer #3 (Remarks to the Author):

This manuscript, which describes a cascade of signaling events required for tail regeneration in larval zebrafish, has been improved by revision. The expanded methods and detailed figure legends describing statistical analyses are particularly appreciated. Additional epistasis experiments also add confidence to the model. The study's novel insights are that ROS produced after injury promote extrusion of the notochord; hedgehog ligands expressed in the extruded notochord bead then activate FGF and Wnt signaling to promote regeneration. These studies raise the question of how ROS promotes notochord extrusion. The experiments here do not answer this question, but it will be an interesting topic for future study, and the authors' speculation about this in the discussion is appreciated.

Minor:

--The new title suggested by the author is an improvement over the previous title

--The paper, while clearly written overall, contains grammatical errors and word omissions in several places (e.g. in the abstract). One particularly confusing instance of this is in Line 180, where I believe the authors meant to say "does not rely", rather than "does rely". The manuscript should be carefully copy edited.

We have reread the manuscript and corrected the typos that we could find.

Note: Major alterations to the text are highlighted in yellow, a separate document that compares the current version with the previous version has also been uploaded.